# Algorithmic Improvements of the KSU-STEM Method Verified on a Fund Portfolio Selection

**Adam Borovička**

Department of Econometrics, University of Economics, 13067 Prague, Czech Republic;
adam.borovicka@vse.cz

**Abstract:** The topic of this article is inspired by the problem faced by many people around the world: investment portfolio selection. Apart from the standardly used methods and approaches, non-traditional multiple objective programming methods can also be significant, providing even more efficient support for making a satisfactory investment decision. A more suitable method for this purpose seems to be a concept working with an interactive procedure through the portfolio that may gradually be adapted to the investor's preferences. Such a method is clearly the Step Method (STEM) or the more suitable improved version KSU-STEM. This method is still burdened by partial algorithmic weaknesses or methodical aspects to think about, but not as much as the other methods. The potentially stronger application power of the KSU-STEM concept motivates its revision. Firstly, an unnecessarily negative principle to determine the basal value of the objectives is revised. Further, the fuzzy goals are specified, which leads to a reformulation of the revealed defuzzified multi-objective model. Finally, the imperfect re-setting of the weights (importance) of unsatisfactory objectives is revealed. Thus, the alternative approaches are proposed. The interventions to the algorithm are empirically verified through a real-life selection of a portfolio of the open unit trusts offered by CONSEQ Investment Management traded on the Czech capital market. This application confirms a significant supporting power of the revised multiple objective programming approach KSU-STEM in a portfolio-making process.

**Keywords:** fuzzy; investment; KSU-STEM; multiple objective; open unit trust; portfolio selection

## 1. Introduction

Real-life decision making can be a very difficult process. Therefore, we should have some supporting tools to make this process easier and more effective. An example of usually a complex decision-making process is investment portfolio selection, which an increasing number of people over the world are facing as they try to valorize their free funds.

To make a portfolio, the intuitive approach, sometimes supported by knowledge of basic quantitative characteristics of the investment (return, risk, etc.), is often applied. Then, the portfolio is especially made on the basis of human intuition, personal mood, or the mood of the crown on the capital market. Such a decision can be supported by a well-known psychological analysis developed by Le Bon [1]. Such an approach is predominantly qualitative. Portfolios can be also made based on fundamental [2] or technical [3] analyses. Then, we receive the value of some quantitative indicator or "graphical" information based on a historical development of the asset prices. Portfolio selection reflecting only some qualitative/quantitative information or one-criterion perspective is too simplifying. Moreover, no mentioned approach enables selecting a portfolio, or exact determination of the assets' shares in the portfolio. They provide only some information on selected investment instruments that support the following steps leading to a portfolio composition making.

To make a more complex and satisfactory investment decision, I propose using a decision-making theory that is often wrongly neglected on the capital market. Of course, a well-known Markowitz optimization model [4] is sometimes applied. However, this model takes into account only one or two characteristics (return and risk). Multiple objective programming methods can reflect many qualitative and quantitative input information on the investment instruments or investor's preferences. Therefore, these multi-criteria approaches are able to provide more complex and representative results. Moreover, they can also work with the outputs of the aforementioned analyses.

In my opinion, the most suitable multiple objective programming methods are interactive approaches reflecting a continuous information from a decision maker (DM). These methods allow a subsequent modification of the actual solution (portfolio) according to the DM's (investor's) preferences. Such an approach helps create a portfolio to the investor's satisfaction. Interactive multiple objective programming methods were developed from the 1970s. The first methods were the Step Method (STEM) designed by Benayoun et al. [5] and the Geoffrion–Dayer–Feinberg's method (GDF) [6]. In the 1980s, a stochastic form of the interactive multi-objective methods was introduced, e.g., [7]. At the same time, a fuzzy form was also proposed, e.g., [8]. Many fuzzy interactive methods work with $\alpha$-cut, e.g., [9]. To complete a brief overview of the interactive procedure, interactive goal programming methods have been also designed, e.g., [10]. Interactive multiple objective programming methods have been evolving for the last few decades; below, we provide a more detailed overview.

Selecting a suitable method is affected by a particular decision-making situation or the DM's abilities. In our case study, a portfolio from open unit trusts offered by CONSEQ Investment Management traded on the Czech capital market is being made. To make a satisfactory investment decision, the following requirements should, in my view, be fulfilled. The algorithm should not require any additional information difficult to determine by the investor (goal, threshold, $\alpha$-cut, distribution of return, explicit relaxation substitution among objective values, etc.). The importance of the objectives must be adjustable by the investor. The strictly determined relaxation of some objectives should be acceptable. The algorithm should be user-friendly for its wider and easy applicability. A method fulfilling all these assumptions can solve the investment problem satisfactorily.

An improved form of extremely popular STEM, KSU-STEM, seems to be a good candidate. Although its algorithm is also not perfect, it has fewer shortcomings compared to other methods. Therefore, this approach deserves further research. In other words, I see current aspects of the algorithm for reflection or improvement. The first partial question is about a determination of the basal (worst) possible value of the objectives. An unnecessary pessimistic approach is modified. Second, the revealed original multi-objective mathematical model is transformed to the one-objective form by introducing a fuzzy goal principle. Such a concept is an efficient alternative to the current formulation. The fuzzy approach can very effectively take into account information about the preferred values of all objective functions (portfolio characteristics) simultaneously. Finally, a recalculation of weights of the unsatisfactory objectives within the interactive procedure is evaluated as redundant. This process can be replaced by simpler ways. The approaches with preserved original weights or with an integration of the weights to the newly determined fuzzy goals are proposed.

The primary aim of this article is to determine a suitable method, more effective than commonly used concepts, as a support for a portfolio selection problem. Based on the algorithmic application abilities, the KSU-STEM approach is chosen as a perfect candidate for further interesting research deepening the ability to solve an investment problem. Therefore, the main aim is to improve its algorithm to solve the problem as satisfactorily as possible. The suitability of these revisions and improvements is tested on a real-life scenario making a portfolio of open unit trusts offered by CONSEQ Investment Management. The more general mission of this paper is to demonstrate a significant application power of (interactive) multiple objective programming methods in the portfolio selection problems in order to use them more in this area.

The structure of the article has the following form. After the introduction, the investment decision-making situation is outlined. The next section discusses the interactive multiple objective programming methods. Then, the algorithm of the STEM and KSU-STEM methods is described. Subsequently, the algorithm is revised using demonstrative investment examples. Finally, a portfolio from the open unit trusts is made by the improved KSU-STEM method. In conclusion, the article is summarized and some ideas for future research are outlined.

## 2. Investment Decision-Making Situation

Recently, investing in open unit trusts has become more and more popular in the Czech Republic. The CONSEQ Investment Management is increasingly gaining ground on the Czech market. This company offers a wide range of open unit trusts, which can be divided into four basic categories: mixed, bond, equity, and real estate funds.

There is no doubt that the essential characteristics of the investment are return and risk. The investment in open unit trusts is also burdened by several fees. In our analysis, the cost is represented by only the initial charge, which reduces the invested amount. All other fees (management, license, etc.) are projected to the fund property or to the fund's return. Other criteria can be a locality of the fund investments, traded currency, style of the fund management, or mood on the capital market. All these aspects can be rather taken into account in a 'preselection' phase when potentially suitable funds for a portfolio are chosen. If the specified client is rather risk-averse (see more below), the risk may be strictly limited. To eliminate the currency risk, funds traded in CZK are chosen. Another important aspect for a fund selection is data availability over the specified time period. For possible type diversification, the funds from all mentioned categories are chosen. Thus, six mixed, four equity, eight bond, and two real estate funds are included, which can be seen in that order with their all characteristics in the following table (Table 1).

**Table 1.** Open unit trusts and their characteristics.

| Fund | Return [%] | Risk [Point] | Cost [%] |
|---|---|---|---|
| Active Invest Dynamic | 0.2083 | 5 | 5 |
| Active Invest Conservative | 0.0697 | 3 | 2.5 |
| Active Invest Balanced | 0.1310 | 4 | 4 |
| Conseq Private Invest Dynamic Portfolio | 0.2809 | 5 | 3 |
| Conseq Private Invest Conservative Portfolio | 0.1112 | 3 | 3 |
| Conseq Private Invest Balanced Portfolio | 0.2032 | 4 | 4 |
| Conseq Invest New Europe Equity A | 0.5914 | 5 | 5 |
| Conseq Invest New Europe Equity B | 0.6236 | 5 | 5 |
| Conseq Invest New Europe Equity D | −0.1009 | 5 | 5 |
| Conseq Opportunity OPFKI | 0.8215 | 7 | 5 |
| Conseq Invest Bond A | 0.0970 | 2 | 2.5 |
| Conseq Invest New Europe Bond A | 0.1042 | 4 | 2.5 |
| Conseq Corporate Bond A | 0.1319 | 2 | 2.5 |
| Conseq Invest Bond B | 0.1211 | 2 | 5 |
| Conseq Invest Bond D | −0.0206 | 2 | 5 |
| Conseq Invest New Europe Bond D | −0.2577 | 4 | 5 |
| Conseq Invest Conservative A | 0.0931 | 2 | 5 |
| Conseq Invest Conservative D | −0.0084 | 2 | 5 |
| Conseq Real Estate | 0.3137 | 2 | 5 |
| Conseq Real Estate Fund | 0.2699 | 6 | 3.5 |

Czech clients are rather conservative, saving over a longer time horizon. Then, the return (in percentage) is calculated as the monthly average from the period from January 2015 to January 2020, which can reflect an actual longer time price development. The risk is measured by a usual indicator, Synthetic Risk and Reward Indicator (SRRI), which is calculated through a standard deviation(s) of returns over the last five years; for more, see the Fund Glossary [11]. Its integer evaluation comes

from the interval $\langle 1,7 \rangle$. The cost is represented by the initial charge as a percentage of the invested amount. Prices and SRRI are taken through the CONSEQ Funds´ List [12]. Returns are calculated in MS Excel.

As mentioned above, an investment strategy that is determined over a longer period of time is the most typical strategy on the Czech capital market with open unit trusts. It usually represents saving money for retirement age or generally for consumption in the more distant future. The purpose of the investment stimulates a conservative approach. Such an investor is more afraid of the investment loss. S/he is willing to lose some part of the return to maintain a lower level of risk. Cost is also considered. However, the initial charge is not as important as the other two characteristics. To weaken a potential loss, the share of equity funds will be limited. For easier portfolio management, the portfolio should contain a reasonable number of assets (three to five funds).

## 3. Decision-Making Theory Approaches for a Portfolio Selection

The potential of the principles of decision-making theory in a portfolio selection is considerable. Therefore, this section focuses on the interactive multiple objective programming methods.

### 3.1. Review of Interactive Multiple Objective Programming Methods

Interactive multiple objective methods have been developing since the 1970s. According to Fiala [13], one of the classification aspects can be a character of the expressed trade-offs: explicit or implicit. One of the oldest methods using explicitly expressed trade-off is called GDF [6]. Explicit trade-off requires an exact quantification of the acceptable relaxation for a couple of the objectives. Some methods (unlike GDF) require this information from the DM or offer these substitution rates to assess by the DM, as in (e.g.,) the Zionts–Wallenius method [14]. Such an active role can be too demanding for the DM. Other methods based on this principle are processed by Miettinen [15]. A more friendly approach in this aspect is represented by the methods using implicit trade-offs. Then, an exact substitution of the objective values is not required. A representative could be the older STEP method (STEM) [5], or younger modification KSU-STEM [16], both using the principle of a minimization of the distance from the ideal solution. Methods based on the reference point have also undergone developments [17]. The aspiration and reservation levels for all objectives must be determined by the DM. This often difficult task tries to be simplified with a "starter" neutral solution. Another method can be Nondifferentiable Interactive Multiobjective Bundle-based Optimization System (NIMBUS), which is based on the 'soft' classification of the objectives into up to five classes; for more, see [18]. Some approaches even use a regression within the interactive procedure, e.g., [19]. Interactive multiple objective methods can also reflect the stochastic elements. Recent development is reflected in modern interactive stochastic approaches [20–22]. A more complex overview of the interactive stochastic multi-objective programming methods can be seen in [23]. To reflect imprecise or uncertain information, the interactive methods with fuzzy elements have been developed [9]. Newer fuzzy methods using mainly $\alpha$-cut principle are designed (e.g.,) in [24]. Finally, a development of goal programming-based methods with an interactive revision of the goals has started in [10]. Recent interactive goal programming methods were proposed (e.g.,) in [25,26].

The selection of a suitable method is namely predetermined by a solved decision-making problem: investment portfolio selecting. This process should be manageable for a wider range of users (investors). Most of them are laics. Making or interactive revision of the portfolio should be instructional. Many methods are burdened by the requirement for additional information from the DM, as mentioned above. It can mean the determination of objective goals (goal programming), thresholds (fuzzy approach), the distribution of stochastic elements (stochastic programming), explicit trade-offs, etc. Such methods are not suitable for our decision-making problem. The method should accept the weights of objective functions because this way of expression of their importance is very friendly thanks to a few easily applicable supporting tools. Finally, the input data or relaxation of the objectives should be possible to set in the easiest way, i.e., in the strict form.

Under these assumptions, many methods are useless (methods working with explicit trade-offs, goals, reference points, fuzzy or stochastic elements, etc.). Some methods are based on a very difficult algorithm (e.g., the Zionts–Wallenius method). Some of them are not even able to work with the weights determined by the DM (e.g., STEM). After such a reduction, the KSU-STEM method seems to be a relevant candidate. Its algorithm is comprehensible. The strict weights determined by the DM are accepted. Only implicit trade-offs are required. On the other side, this algorithm is not fully perfect. In other words, some fragments of the algorithm stimulate subjective thinking.

### 3.2. STEM and KSU-STEM

At first, the STEM and KSU-STEM algorithms are described. Then, both algorithms are compared. Their drawbacks or shortcomings are reflected.

### 3.2.1. STEM Algorithm

The STEM algorithm can be described, as by Benayoun et al. [5], in the following several steps.

Step 1: Define $k$ objective functions (objectives) $f_1(\mathbf{x}), f_2(\mathbf{x}), \ldots, f_k(\mathbf{x})$, where $\mathbf{x} = (x_1, x_2, \ldots, x_n)^T$ is a vector of $n$ variables. Distinguish the set of indices of the minimizing or maximizing objective functions denoted as $J_{min}$ or $J_{max}$. Further, the set including all conditions of the solved problem is denoted as $X^1$. Then, the ideal value of each objective function can be found on this set. Let us denote the optimal value of the $j$-th minimizing or maximizing objective function as $f_j^I = f_j(\mathbf{x_j^*})$, when the following holds

$$\begin{aligned} \mathbf{x_j^*} = \arg\min f_j(\mathbf{x}) \\ \mathbf{x} \in X^1 \end{aligned} \quad j \in J_{min}, \qquad \begin{aligned} \mathbf{x_j^*} = \arg\max f_j(\mathbf{x}) \\ \mathbf{x} \in X^1 \end{aligned} \quad j \in J_{max}. \tag{1}$$

Let $\mathbf{Z} = (z_{ij})$ be the matrix with the generic elements

$$z_{ij} = f_j(\mathbf{x_i^*}) \qquad i, j = 1, 2, \ldots, k, \text{ or} \tag{2}$$
$$z_{ij} = f_i^I = f_j^I \qquad i = j,$$

where $\mathbf{x_i^*}$ is consistent with (1) when $i = j$. Let $\mathbf{w} = (w_1, w_2, \ldots, w_k)^T$ be a vector of the weights of objectives. The weight of the $j$-th objective is calculated as follows

$$\begin{aligned} w_j = \frac{\max\limits_{1 \le i \le k} z_{ij} - f_j^I}{\max\limits_{1 \le i \le k} z_{ij}} \alpha \left( \sqrt{\sum_{p=1}^n c_{jp}^2} \right)^{-1} \qquad j \in J_{min} \\ \\ w_j = \frac{f_j^I - \min\limits_{1 \le i \le k} z_{ij}}{f_j^I} \alpha \left( \sqrt{\sum_{p=1}^n c_{jp}^2} \right)^{-1} \qquad j \in J_{max} \end{aligned}, \tag{3}$$

where $c_{jp}, j = 1, 2, \ldots, k, p = 1, 2, \ldots, n$, is a coefficient by the $p$-th variable in the $j$-th objective function. The value $\alpha$ is set so that the sum of the weights is one. A weight calculation was variously modified by Eschenauer et al. [27] or Vanderpooten and Vincke [28].

Step 2: In the next step, the weighted Chebyshev problem should be solved

$$\begin{aligned} \min \max\limits_{1 \le j \le k} \left( w_j \left| f_j^I - f_j(\mathbf{x}) \right| \right) \\ \mathbf{x} \in X^1 \end{aligned}. \tag{4}$$

This model can be easily linearized through the *minmax* optimization approach as follows

$$\min \quad D$$

$$w_j \left( f_j(\mathbf{x}) - f_j^I \right) \le D \qquad j \in J_{min}$$
$$w_j \left( f_j^I - f_j(\mathbf{x}) \right) \le D \qquad j \in J_{max} \tag{5}$$

$$\mathbf{x} \in X^1$$

Let $\mathbf{x^1}$ be a vector representing an optimal solution of the Model (5) with the objective values $f_j(\mathbf{x^1}), j = 1, 2, ..., k.$ If the values of all the objectives are acceptable for a decision maker (investor), the compromise solution, or satisfactory portfolio composition, is found. The process is terminated. On the other side, if the values of all objectives are not acceptable, the algorithm is also finished. It is not possible to improve simultaneously the values of all objectives. Thus, solution $\mathbf{x^1}$ is non-dominated. The improvement of one objective value is conditioned by a relaxation of at least one other objective value. If it is acceptable for a decision maker, the interactive procedure finding a compromise solution can begin.

Step 3: Let us denote the set $J_{min}^R$ or $J_{max}^R$ including the indices of the minimizing or maximizing objective functions with a satisfactory value. Then, a decision maker provides $\Delta f_j(\mathbf{x^1}), j \in J_{min}^R \cup J_{max}^R$, as the amount of acceptable relaxation for the $j$-th minimizing or maximizing objective function. Then, the set $X^2$ including the following conditions is specified

$$
\begin{aligned}
f_j(\mathbf{x}) &\le f_j(\mathbf{x^1}) & j \in J_{min} - J_{min}^R \\
f_j(\mathbf{x}) &\ge f_j(\mathbf{x^1}) & j \in J_{max} - J_{max}^R \\
f_j(\mathbf{x}) &\le f_j(\mathbf{x^1}) + \Delta f_j(\mathbf{x^1}) & j \in J_{min}^R \\
f_j(\mathbf{x}) &\ge f_j(\mathbf{x^1}) - \Delta f_j(\mathbf{x^1}) & j \in J_{max}^R
\end{aligned}
\tag{6}
$$

where $\Delta f_j(\mathbf{x^1}) \ge 0, j \in J_{min}^R \cup J_{max}^R.$ Now, a new distance Chebyshev problem, or its linearized form can be formulated to find another (better) solution

$$\min \quad D$$

$$\min \max_{1 \le j \le k} \left( w_j' \left| f_j^I - f_j(\mathbf{x}) \right| \right) \qquad w_j' \left( f_j(\mathbf{x}) - f_j^I \right) \le D \qquad j \in J_{min}$$
$$\text{, or} \qquad w_j' \left( f_j^I - f_j(\mathbf{x}) \right) \le D \qquad j \in J_{max} \tag{7}$$
$$\mathbf{x} \in X^1 \cup X^2 \qquad \qquad \mathbf{x} \in X^1 \cup X^2$$

where

$$
\begin{aligned}
w_j' &= 0 & j \in J_{min}^R \cup J_{max}^R \\
w_j' &= w_j & j \in J_{min} - J_{min}^R, j \in J_{max} - J_{max}^R
\end{aligned}
\tag{8}
$$

The optimal solution $\mathbf{x^2}$ of Model (7) is found. If the solution is acceptable, the compromise solution (satisfactory portfolio composition) is found. The process is terminated. If not, the interactive procedure continues until a decision maker accepts the solution.

Models (1), (5), or (7) are solvable because of the limited (non-empty) set of feasible solutions. A selection of the method solving these problems is based on the (non)linearity of the functions and integer conditions. After that, a global or local optimum is found.

### 3.2.2. KSU-STEM Algorithm

The algorithm is described step by step according to Lai and Hwang [16]. Of course, some parts of the KSU-STEM algorithm are the same as the STEM algorithm. A notation in the description of both algorithms is consistent.

Step 1: Define $k$ objective functions (objectives) $f_1(\mathbf{x}), f_2(\mathbf{x}), ..., f_k(\mathbf{x}),$ where $\mathbf{x} = (x_1, x_2, ..., x_n)^T$ is a vector of $n$ variables. Distinguish the set of indices of the minimizing or maximizing objective

functions denoted as $J_{min}$ or $J_{max}$. The importance of the criteria is expressed by the vector of weights $\mathbf{w} = (w_1, w_2, \ldots, w_k)^T$, where $w_j, j = 1, 2, \ldots, k,$ is the weight of the *j*-th objective function. Many methods for weight estimation are known, such as the scoring metod, Saaty's or Fuller's approach; for more, see [29,30]. For simple user-friendly implementation, a scoring method is recommended. Then, let us specify a scoring interval $\langle 1, 10 \rangle$ from which a decision maker assigns the score for each objective according to his/her preferences. The lowest or highest score represents the weakest, strongest preference, or objective importance. The weight of the *j*-th objective function can be calculated as follows

$$w_j = \frac{s_j}{\sum_{j=1}^{k} s_j} \qquad j = 1, 2, \ldots, k,$$

(9)

where $s_j, j = 1, 2, \ldots, k,$ is a score assigned to the *j*-th objective function. It is obvious that the weights are traditionally in a standardized form, which is more suitable for practical use. Thus, $\sum_{j=1}^{k} w_j = 1$ holds.

Step 2: The minimum and maximum of each *j*-th objective function are determined on the set of feasible solutions $X^1$ (containing all conditions of the solved problem) by finding a solution of the following models

$$\min_{\mathbf{x} \in X^1} f_j(\mathbf{x}), \qquad \max_{\mathbf{x} \in X^1} f_j(\mathbf{x}).$$

(10)

The solutions of these models represent ideal $f_j^I$ (the lowest or highest value for minimizing or maximizing the objective function) and basal value $f_j^B$ (the highest or lowest value for minimizing or maximizing the objective function) of each *j*-th objective function.

Step 3: The mathematical model minimizing a maximal weighted relative (standardized) deviation from the ideal solution (value) is formulated as follows

$$\min \alpha$$
$$w_j \frac{f_j(\mathbf{x}) - f_j^I}{f_j^B - f_j^I} \leq \alpha \qquad j \in J_{min}$$
$$w_j \frac{f_j^I - f_j(\mathbf{x})}{f_j^I - f_j^B} \leq \alpha \qquad j \in J_{max}.$$
$$\mathbf{x} \in X^1$$
$$0 \leq \alpha \leq 1$$

(11)

The optimal solution is denoted as $\mathbf{x}^1$ with the values of all *k* objective functions $f_j(\mathbf{x}^1), j = 1, 2, \ldots, k.$ If the values of all objective functions are acceptable by a decision maker, the compromise solution is found. If the values of all objective function are unacceptable, the algorithm is also terminated. It is not possible to simultaneously improve all objective values because the solution $\mathbf{x}^1$ is non-dominated. If the values of some objectives are acceptable and some are not, then the interactive procedure can be started to reveal the compromise solution.

Step 4: To improve the values of unsatisfactory objectives, a decision maker must relax at least one satisfactory objective. Similar to the STEM algorithm, let $J_{min}^R$ or $J_{max}^R$ be the set containing the indices of the minimizing or maximizing objective functions with a satisfactory value. Then, a decision maker provides $\Delta f_j(\mathbf{x}^1), j \in J_{min}^R,$ or $\Delta f_j(\mathbf{x}^1), j \in J_{max}^R,$ as the amount of acceptable relaxation for the *j*-th minimizing or maximizing objective. Then, the set $X^2$ including the following conditions is specified in this form

$$
\begin{aligned}
f_j(\mathbf{x}) \leq f_j(\mathbf{x^1}) && j \in J_{min} - J_{min}^R \\
f_j(\mathbf{x}) \geq f_j(\mathbf{x^1}) && j \in J_{max} - J_{max}^R \\
f_j(\mathbf{x}) \leq f_j(\mathbf{x^1}) + \Delta f_j(\mathbf{x^1}) && j \in J_{min}^R \\
f_j(\mathbf{x}) \geq f_j(\mathbf{x^1}) - \Delta f_j(\mathbf{x^1}) && j \in J_{max}^R
\end{aligned}
\tag{12}
$$

Finally, the weights are reformulated as

$$
\begin{aligned}
w'_j &= 0 && j \in J_{min}^R \cup J_{max}^R \\
w'_j &= \frac{w_j}{\displaystyle\sum_{\substack{j=1 \\ j \notin J_{min}^R \cup J_{max}^R}}^{k} w_j} && j \in J_{min} - J_{min}^R, j \in J_{max} - J_{max}^R
\end{aligned}
\tag{13}
$$

The following modified Model (11) is solved

$$
\begin{aligned}
& \min \ \alpha \\
& w'_j \frac{f_j(\mathbf{x}) - f_j^I}{f_j^B - f_j^I} \leq \alpha && j \in J_{min} - J_{min}^R \\
& w'_j \frac{f_j^I - f_j(\mathbf{x})}{f_j^I - f_j^B} \leq \alpha && j \in J_{max} - J_{max}^R \\
& \mathbf{x} \in X^1 \cup X^2 \\
& 0 \leq \alpha \leq 1
\end{aligned}
\tag{14}
$$

The optimal solution $\mathbf{x^2}$ of Model (14) is another compromise solution. If this solution is still not acceptable, the improvement procedure is repeated. Otherwise, the compromise solution is found, and the algorithm is over.

The solvability of all models is the same in terms of the STEM approach. They can be solved using a standard system to support modeling such as LINGO, MPL for Windows, etc.

### 3.2.3. STEM vs. KSU-STEM

After a proper introduction to the algorithms, their effective comparison can be made.

1.  The first KSU-STEM positive is a possibility of determination of the weights of objectives by the DM. On the other side, this fact can also be a disadvantage for the DMs who are not able to determine the weights. As there are many supportive tools for weight estimation, this will reflect a minority of cases.
2.  Moreover, a calculation of STEM weights may not potentially work properly if the objective values are negative.
3.  In the original form of the STEM approach, the distances are not standardized, which can distort the result. On the contrary, KSU-STEM applies a relative standardized distance, which makes the result more reliable.
4.  KSU-STEM works better with a combination of minimizing and maximizing objective functions. Any transformation of the objective character is not required.
5.  The ideal value is determined via the same approach. However, KSU-STEM determines the basal value at an unnecessarily pessimistic level. This aspect can be considered as a minor drawback of KSU-STEM. This shortcoming is eliminated in the revised KSU-STEM described below.
6.  On the other side, these two extreme values of the objectives are artificially used to represent the normalization of their values. STEM uses only the ideal value in this process.
7.  Both methods zero the weights of satisfactory objectives. KSU-STEM, compared to STEM, recalculates the weights of other objectives, which is actually necessary action. This fact must also be duly examined (see more below).

The revised KSU-STEM algorithm below reflects the positive aspects of the original KSU-STEM algorithm and corrects its above-mentioned partial shortcomings.

## 4. Improvements of the KSU-STEM Algorithm

The main part of the article introduces the possible improvements, or modifications, of the KSU-STEM algorithm that further enhance the already great application (and theoretical) power. Let us proceed step by step through the algorithm.

### 4.1. Step 2: Basal and Ideal Value of the Objectives

The way for a determination of the objective ideal value is meaningful. It is a minimum $f_j(\mathbf{x}_j^*) = f_j^I, j \in J_{min}$, or maximum $f_j(\mathbf{x}_j^*) = f_j^I, j \in J_{max}$, of the *j*-th minimizing or maximizing objective function on the set of feasible solutions. Thus, the following holds for the *j*-th minimizing or maximizing objective function:

$$\mathbf{x}_j^* = \arg\min_{\mathbf{x} \in X^1} f_j(\mathbf{x}), \text{ or } \quad \mathbf{x}_j^* = \arg\max_{\mathbf{x} \in X^1} f_j(\mathbf{x}). \tag{15}$$

Models are solvable because the set $X^1$ is bounded in the portfolio-making problem (at least) due to the 'portfolio' condition (unit sum of the assets' shares). Of course, some objective function(s) can be nonlinear. Then, only a local optimum can be found.

The opposite extremes for all objectives are used for a determination of the basal (worst) value. This concept is unnecessarily pessimistic. A more reasonable concept determines the basal value of the objective function with respect to the optimal solution for other objective functions. This concept is actually integrated in the STEM algorithm. So, the basal value of the *j*-th minimizing or maximizing objective function is computed via the following formula

$$f_j^B = \max_{\mathbf{x}_j^*}\left[ f_j(\mathbf{x}_j^*) \right] \qquad j \in J_{min}, \text{ or}$$
$$f_j^B = \min_{\mathbf{x}_j^*}\left[ f_j(\mathbf{x}_j^*) \right] \qquad j \in J_{max}. \tag{16}$$

The different approach can be practically illustrated on our investment portfolio-selecting problem. The basal values via KSU-STEM and the improved approach are presented in Table 2.

**Table 2.** Basal values of the objectives via both approaches.

| Approach | Return | Risk | Cost |
|---|---|---|---|
| KSU-STEM | −0.136 | 5.9 | 5 |
| Improved approach | 0.029 | 4.3 | 4.3 |

It confirms that the result of the KSU-STEM procedure provides an unnecessarily negative result compared with the improved approach. Such bad values have never been gained by the objectives.

### 4.2. Step 3: Fuzzy Goal Construction

In the third step, a one-objective model (11) is solved. This model is based on a minimization of the maximal weighted relative deviation from the ideal solution. Although the authors of KSU-STEM do not explicitly mention it, this model may be a transformation of the following multi-objective model

$$\begin{aligned} &\min \ f_j(\mathbf{x}) \qquad j \in J_{min} \\ &\max \ f_j(\mathbf{x}) \qquad j \in J_{max}. \\ &\mathbf{x} \in X^1 \end{aligned} \tag{17}$$

The used principle is derived from the STEM approach, which minimizes a deviation from the ideal solution measured by the Chebyshev metric. Then, a linearized one-objective model leads to a *minmax* optimization problem.

In my opinion, another (alternative) concept for finding a solution of Model (17) should be developed. This principle is actually the opposite of the original form. It represents a positive view: the maximization of 'something'. Specifically, the approach of a minimization of a maximum

weighted relative distance from the ideal solution is transformed to the approach of a maximization of the minimum weighted relative distance from the basal value. This transformation is clearly performed using fuzzy programming principles. Fuzzy sets, respectively triangular fuzzy numbers, make it possible to express quantitatively the preference of acquiring a specific value of the objective function through the linear relation between the basal and ideal value. Then, a fuzzy multiple objective programming model can reflect the "value" preferences of all objective functions simultaneously. To understand this part properly, the basics of fuzzy set theory and fuzzy optimization principles should be introduced (for more, see the Appendix). In the first step, the *fuzzy goals* must be specified. Then, the *j*-th minimizing objective function, $f_j(\mathbf{x}), j \in J_{min}$, is actually transformed to the fuzzy goal represented by the fuzzy set $\tilde{F}_{f_j(\mathbf{x})}, j \in J_{min}$, with the following membership function

$$\mu_{\tilde{F}_{f_j(\mathbf{x})}}[f_j(\mathbf{x})] = \begin{cases} 1 & f_j(\mathbf{x}) \leq f_j^I \\ \frac{f_j^B - f_j(\mathbf{x})}{f_j^B - f_j^I} & f_j^I \leq f_j(\mathbf{x}) \leq f_j^B \\ 0 & f_j(\mathbf{x}) \geq f_j^B \end{cases} . \tag{18}$$

The *j*-th maximizing objective function is also transformed to the fuzzy goal as the fuzzy set $\tilde{F}_{f_j(\mathbf{x})}, j \in J_{max}$, with the membership function

$$\mu_{\tilde{F}_{f_j(\mathbf{x})}}[f_j(\mathbf{x})] = \begin{cases} 1 & f_j(\mathbf{x}) \geq f_j^I \\ \frac{f_j(\mathbf{x}) - f_j^B}{f_j^I - f_j^B} & f_j^B \leq f_j(\mathbf{x}) \leq f_j^I \\ 0 & f_j(\mathbf{x}) \leq f_j^B \end{cases} . \tag{19}$$

The membership functions (18) and (19) can be graphically displayed as follows (Figure 1).

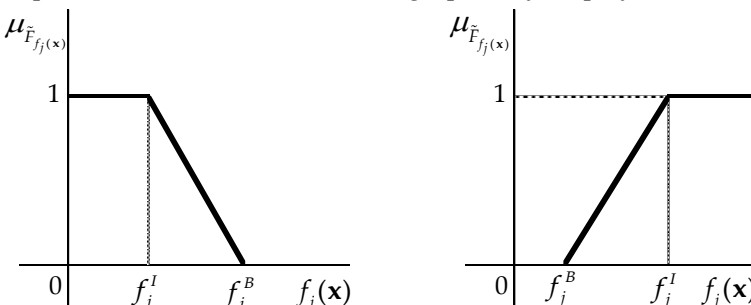

**Figure 1.** Membership functions of minimizing (**left**) and maximizing (**right**) fuzzy goals.

These are actually right and left-hand triangular fuzzy numbers. This type of fuzzy number is selected because its membership function is piecewise linear. Therefore, working with them is easier. In addition, the shape of the membership function enables reflecting the relationship between the objective value and the membership grade well. Thanks to Bellman's fuzzy goal optimization principle [31], the multiple objective model shown in (17) can be transformed to a one-objective form as follows

$$\max \ \alpha$$

$$(1 - w_j) \frac{f_j^B - f_j(\mathbf{x})}{f_j^B - f_j^I} \geq \alpha \qquad j \in J_{min}$$

$$(1 - w_j) \frac{f_j(\mathbf{x}) - f_j^B}{f_j^I - f_j^B} \geq \alpha \qquad j \in J_{max} \tag{20}$$

$$\mathbf{x} \in X^1$$

$$0 \leq \alpha \leq 1$$

where $\alpha$ is an intersection of all the fuzzy sets in Model (20) representing the (weighted) membership grade of a solution. Therefore, $\alpha$ is maximized. A real (not weighted) membership grade of solution is as follows:

$$\min_{1 \le j \le k} \left( \frac{f_j^B - f_j(\mathbf{x})}{f_j^B - f_j^I}, \frac{f_j(\mathbf{x}) - f_j^B}{f_j^I - f_j^B} \right). \tag{21}$$

Let us get back to the weights in Model (20). As we can see, the position of the weights is different compared to Model (11) to reflect the importance of the objectives correctly. Of course, the same principle of a weight position is used in Model (24) within an interactive procedure. Another way of calculating an importance expression is to divide the left side of the conditions (related to the fuzzy goal) by the weight. However, it is an alternative approach that requires an elimination of the condition for the $\alpha$ value, because $\alpha$ could then be greater than 1. Then, Model (20) is transformed to the following form

$$
\begin{aligned}
&\max \ \alpha \\
&\frac{f_j^B - f_j(\mathbf{x})}{w_j(f_j^B - f_j^I)} \ge \alpha \qquad j \in J_{min} \\
&\frac{f_j(\mathbf{x}) - f_j^B}{w_j(f_j^I - f_j^B)} \ge \alpha \qquad j \in J_{max} \ . \\
&\mathbf{x} \in X^1 \\
&\alpha \ge 0
\end{aligned}
\tag{22}
$$

The elimination of the condition reflecting the upper bound for $\alpha$ seems to be a sensible idea in our portfolio selection case. Of course, the original interpretation of the $\alpha$ values is not maintained. However, its "technical" function in the model is the same. In the following table, we can see the significant difference in portfolio characteristics. It is obvious from Table 3 that Model (22) revised by the condition $\alpha \le 1$ would not provide the results in accordance with the expressed preferences of risk-averse investor specified above. Of course, a composition of the portfolio is significantly different.

**Table 3.** Characteristics of the portfolio selected via Model (22) and its revised form.

| Model (22) | Return | Risk | Cost |
|:---:|:---:|:---:|:---:|
| Original | 0.252 | 2.45 | 3.977 |
| With $\alpha \le 1$ | 0.17 | 3.083 | 3.744 |

The conditions $0 \le \alpha \le 1$ in Model (20) are actually necessary because the $\alpha$ value is always in this interval.

### 4.3. Step 4: Weight Recalculation

Another revision of the KSU-STEM algorithm deals with a recalculation of the weights within the interactive procedure. Within the KSU-STEM and STEM algorithms, the weights of satisfactory objectives become zero. The process of recalculation of the weights of unsatisfactory objectives is different. The STEM algorithm leaves the weights in the original form. In KSU-STEM, the weights are recalculated to the standardized form. In other words, the sum of the weights is 1.

#### 4.3.1. Special Case of Single Unsatisfactory Objective

At first, a special case of one unsatisfactory objective should be introduced. In the original KSU-STEM algorithm, a unit weight of the unsatisfactory objective works well in Model (11). The problem occurs in the proposed modified Model (20) because zero is on the left side of the condition, and it is related to the fuzzy goal. The fuzzy goal is virtually eliminated, which potentially distort the results. The DM's preferences are not correctly taken into account. In such a situation, the following

modification is proposed. The weight is transformed to the form of 'almost one'. Then, one is reduced by the infinitesimal constant as follows

$$w_j = 1 - 10^{-8} \qquad j \in J^S, \tag{23}$$

where $J^S = \left\{ J_{min} - J_{min}^R \right\} \cup \left\{ J_{max} - J_{max}^R \right\}$ is the single element set containing the index of one unsatisfactory objective.

### 4.3.2. Necessity of the Weight Recalculation?

The main question is about the need to recalculate the weights. Of course, the recalculation does not change the relative relations among the weights. On the other side, the absolute differences are naturally modified. Given again our investment portfolio selecting problem, after all conditions (see below in more detail), the initial portfolio has the following form: 40% *Conseq Corporate Bond A*, 25.03% *Conseq Real Estate*, 17.87% *Conseq Invest Bond A,* and 17.1% *Conseq Invest Europe Equity B*. The portfolio return is 0.255%, the risk is 2.513, and the cost is 3.553%.

Under the risk-averse strategy and knowledge of the extreme values of the objectives, the investor is not satisfied with the level of portfolio risk. To reduce this value, the investor is able to accept a cost increase under the same portfolio return if possible. Thus, the cost can be increased by 0.147 percentage points (up to 3.7%). Via the revised KSU-STEM, the weight of risk is determined by Model (23). The weights of the other two objectives are zero. The portfolio composition is: 37.32% *Conseq Corporate Bond A*, 32.68% *Conseq Real Estate*, 15% *Conseq Invest Bond A*, and 15% *Conseq Invest Europe Equity B*. The return of the portfolio is the same (0.255%), the risk is 2.45, and the cost is 3.692%.

If the STEM concept for a weight 'recalculation' is used (original weight of risk, zero weights of return and cost), the result is the same. Other combinations of the objectives for their value improvement were selected (e.g., return and cost, risk and cost). Then, a composition of the portfolios is also the same by means of both approaches. The main reason is that the relative relations among the weights of unsatisfactory objectives remain the same.

After all previous revisions, the model for a solution improvement in terms of the interactive procedure is formulated as

$$\max \ \alpha$$

$$(1 - w_j) \frac{f_j^B - f_j(\mathbf{x})}{f_j^B - f_j^I} \geq \alpha \qquad j \in J_{min} - J_{min}^R$$

$$(1 - w_j) \frac{f_j(\mathbf{x}) - f_j^B}{f_j^I - f_j^B} \geq \alpha \qquad j \in J_{max} - J_{max}^R \ . \tag{24}$$

$$\mathbf{x} \in X^1 \cup X^2$$

$$0 \leq \alpha \leq 1$$

The set of conditions $X^2$ formulated within the interactive procedure should be also discussed. The original formulation (12) is not perfect because it also enables a non-improvement unsatisfactory objective thanks to the following expression

$$f_j(\mathbf{x}) \leq f_j(\mathbf{x}^1) \qquad j \in J_{min} - J_{min}^R$$
$$f_j(\mathbf{x}) \geq f_j(\mathbf{x}^1) \qquad j \in J_{max} - J_{max}^R \ . \tag{25}$$

Mainly in the case of a higher number of unsatisfactory objectives, the conditions should be transformed to the sharp inequalities

$$f_j(\mathbf{x}) < f_j(\mathbf{x}^*) \qquad j \in J_{min} - J_{min}^R$$
$$f_j(\mathbf{x}) > f_j(\mathbf{x}^*) \qquad j \in J_{max} - J_{max}^R \ . \tag{26}$$

Sharp inequalities (26) may be problematic when searching for a solution of the models. Then, the conditions can be transformed via the infinitesimal constant as follows

$$f_j(\mathbf{x}) \leq f_j(\mathbf{x}^*) - 10^{-8} \quad j \in J_{min} - J_{min}^R$$
$$f_j(\mathbf{x}) \geq f_j(\mathbf{x}^*) + 10^{-8} \quad j \in J_{max} - J_{max}^R$$

(27)

This formulation holds only under the assumption that the sets $J_{min}^R$ and $J_{max}^R$ contain the indices of all satisfactory objectives, including cases with a zero relaxation value. Although the authors do not explicitly declare this fact, I consider it reasonable. If not, the weights of the satisfactory objective would be also taken into account in the non-zero form. Then, the conditions/fuzzy goals with these weights would not be eliminated. The result could be distorted. A similar situation could arise by maintaining the non-zero weights of the satisfactory objectives with a particular acceptable relaxation level. An explicitly expressed preference (by means of the weights or relevant conditions) about a possible relaxation can inadequately 'muffle' this allowed relaxation against the unsatisfactory objectives. Then, $\alpha$ can be improperly reduced. Our investment situation confirms this declaration. After a formulation of the additional preferences of a portfolio improvement mentioned above, a new portfolio, obtained as described, has the return at the level of 0.262%, a risk of 2.462, and a cost of 3.7%. As we can see, the most important risk is a little worse (compared to 2.45). The cost is also at a worse level (compared to 3.692%). The return is slightly better (compared to 0.255%). Under the additional preferences, this solution is unnecessarily bad.

Thus, the sets $J_{min}^R$ and $J_{max}^R$ must contain all satisfactory objectives. Then, the revised Model (24) with the set $X^2$ modified by Model (27) reflects the DM's requirements for a solution improvement. The weights of unsatisfactory objectives are explicitly included. The weights (importance) of satisfactory objectives are presented implicitly through the minimum or maximum restrictive value in the revised set $X^2$.

### 4.3.3. Fuzzy Goal Modification

In terms of the (revised) KSU-STEM interactive procedure, the additional preferences, or the modified importance (weight) of the objectives can be reflected by another way (besides the two approaches described above). I propose a modification of the established fuzzy goals for the unsatisfactory objectives. A stronger preference can be reflected by the improvement of the basal value. It means its decrease or increase for a minimizing or maximizing objective. A modification of the fuzzy goals can be graphically displayed as follows (Figure 2).

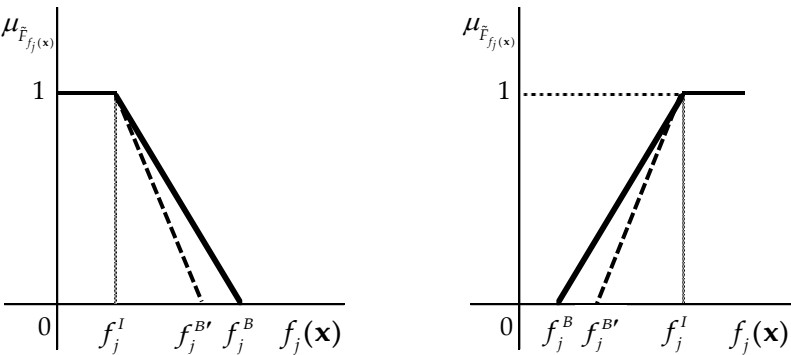

**Figure 2.** Modification of minimizing (**left**) and maximizing (**right**) fuzzy goals.

It is obvious that a modified basal value pushes to improve the value of the *j*-th objective function $f_j(\mathbf{x})$. The main question is how to get a new basal value $f_j^{B'}$. I propose an approach based on the integration of the weight to the fuzzy goal or (triangular) fuzzy number. Then, the following formula holds for the *j*-th minimizing or maximizing fuzzy goal

$$f_j^{B'} = f_j^B - w_j \left( f_j^B - f_j^I \right) \quad j \in J_{min} - J_{min}^R \text{, or}$$
$$f_j^{B'} = f_j^B + w_j \left( f_j^I - f_j^B \right) \quad j \in J_{max} - J_{max}^R.$$

(28)

Then, the modified fuzzy goals are in the following form

$$\frac{f_j^{B'} - f_j(\mathbf{x})}{f_j^{B'} - f_j^{I}} \geq \alpha \qquad j \in J_{min} - J_{min}^{R}$$

$$\frac{f_j(\mathbf{x}) - f_j^{B'}}{f_j^{I} - f_j^{B'}} \geq \alpha \qquad j \in J_{max} - J_{max}^{R}$$

(29)

Another part of Model (24) used in the interactive procedure remains unchanged. In terms of the interactive procedure of selecting the investment portfolio, the modified basal value of the risk $f_l^{B'}$ is computed as follows

$$f_2^{B'} = 4.3 - 0.529(4.3 - 2) = 3.083.$$

(30)

Then, the fuzzy goal of the portfolio risk is modified to the following form

$$\frac{3.083 - \sum_{i=1}^{20} l_i x_i}{3.083 - 2} \geq \alpha.$$

(31)

Model (24) with the modified fuzzy goal for a risk shown in Equation (31) is solved to select the revised portfolio. The result (portfolio composition) is the same. The same situation also occurs for other combinations of improved objectives (return and cost or risk and cost). So, this approach can be comprehended as an alternative to the approach with the explicitly expressed weights of unsatisfactory objectives.

Now, we have actually three alternative approaches to take into account the preferences of an unsatisfactory objective value(s): the original KSU-STEM with a weight recalculation, a revised approach with the original weight of an unsatisfactory objective, and a revised approach of modified fuzzy goals via the integrated weights. The second approach has one advantage compared to two others. It is easier to apply thanks to actually having no need to recalculate the weights or basal values. It may be that a formulation of the fuzzy goal is simpler because of the integrated weights. However, the revised approach with the original weight is clearly the most user-friendly.

## 5. Selecting a Portfolio of CONSEQ Funds via Improved KSU-STEM

Finally, a revised version of KSU-STEM is applied to make a portfolio of the open unit trusts offered by CONSEQ Investment Management. As mentioned above, it is a longer-time risk-averse investment. The weights of three determined criteria (objectives) are calculated via a scoring method. The scores are assigned according to the preference of the risk-averse investment strategy specified in Section 2. The scores and weights are in the following table (Table 4).

**Table 4.** Scores and weights of the objectives.

| Objective | Score | Weight |
|-----------|-------|--------|
| Return | 6 | 0.353 |
| Risk | 9 | 0.529 |
| Cost | 2 | 0.118 |

For easier portfolio management, the number of open unit trusts is limited by minimum (15%) and maximum (40%) share in the portfolio. Further, the risk-averse investor limits a share of equity funds to 25% because these funds potentially generate the highest loss. All strategy aspects can be determined by a "more advanced" investor or with the assistance of an experienced investment counsel. The same applies to the application of the method itself.

The objective functions representing three portfolio characteristics are formulated in the weighted average form as follows

$$f_1(\mathbf{x}) = \sum_{i=1}^{20} r_i x_i \text{ - portfolio return,}$$

(32)

$$f_2(\mathbf{x}) = \sum_{i=1}^{20} l_i x_i \text{ - portfolio risk,}$$

$$f_3(\mathbf{x}) = \sum_{i=1}^{20} c_i x_i \text{ - portfolio cost,}$$

where $r_i, l_i, c_i, i = 1, 2, ..., 20,$ represent the return, risk, and cost of the *i*-th open unit trust (indexed in order from Table 1) and $x_i, i = 1, 2, ..., 20,$ represents a share of the *i*-th fund in the portfolio generating the *n*-component vector of variables x. The ideal (optimal) value of each objective is determined through Model (10) in the following form

$$\begin{array}{ccc} \max \; \displaystyle\sum_{i=1}^{20} r_i x_i & \min \; \displaystyle\sum_{i=1}^{20} l_i x_i & \min \; \displaystyle\sum_{i=1}^{20} c_i x_i \\ x_i \in X^1 \quad i = 1, 2, ..., 20 & x_i \in X^1 \quad i = 1, 2, ..., 20 & x_i \in X^1 \quad i = 1, 2, ..., 20 \end{array}, \quad (33)$$

where the feasible solution set $X^1$ includes the following conditions

$$0.15 y_i \le x_i \le 0.4 y_i \quad i = 1, 2, ..., 20 \tag{34}$$

$$\sum_{i=7}^{10} x_i \le 0.25 \tag{35}$$

$$\sum_{i=1}^{20} x_i = 1 \tag{36}$$

$$x_i \ge 0 \quad i = 1, 2, ..., 20 \tag{37}$$

$$y_i \in \{0, 1\} \quad i = 1, 2, ..., 20 \tag{38}$$

where the binary variable $y_i, i = 1, 2, ..., 20,$ helps to ensure, within a set of constraints (34), a fixed possible interval for the share of each *i*-th open unit trust in the portfolio. The second condition (35) represents the limit of equity fund share in the portfolio. Further, there is a standard condition for making a portfolio (36). Non-negativity and binarity conditions (37), (38) for the relevant variables must not be missed.

The basal values of the objectives are determined via the revised concept in Model (16). Then, both extremes of the objectives can be displayed in the following table (Table 5).

**Table 5.** Extremes of the objectives.

| Objective | Basal | Ideal |
|-----------|-------|-------|
| Return | 0.029 | 0.429 |
| Risk | 4.3 | 2 |
| Cost | 4.3 | 2.5 |

Now, the initial investment portfolio is made via the multi-objective model (17)

$$\begin{array}{c} \min \; \displaystyle\sum_{i=1}^{20} l_i x_i \\ \min \; \displaystyle\sum_{i=1}^{20} c_i x_i \\ \max \; \displaystyle\sum_{i=1}^{20} r_i x_i \\ x_i \in X^1 \quad i = 1, 2, ..., 20 \end{array}, \quad (39)$$

which can be transformed to the maxmin form (20) in the particular shape

$$\max \quad \alpha$$

$$(1-0.529)\frac{4.3-\sum_{i=1}^{20}l_i x_i}{2.3} \geq \alpha$$

$$(1-0.118)\frac{4.3-\sum_{i=1}^{20}c_i x_i}{1.8} \geq \alpha \quad .$$

$$(1-0.353)\frac{\sum_{i=1}^{20}r_i x_i - 0.029}{0.4} \geq \alpha$$

$$x_i \in X \quad i = 1,2,...,20$$

$$0 \leq \alpha \leq 1 \tag{40}$$

Mathematical models (33) and (40) are solved in the LINGO optimization software. The portfolio (as the optimal solution of (40)) has the following form (as already mentioned above): 40% *Conseq Corporate Bond A*, 25.03% *Conseq Real Estate*, 17.87% *Conseq Invest Bond A,* and 17.1% *Conseq Invest Europe Equity B*. The portfolio return (average monthly return) is 0.255%, the risk (measured as SRRI) is 2.513, and the cost (entry fee) is 3.553%. It is not surprising that the fund with the lowest level of risk dominates in the portfolio. The presence of a higher-risky equity fund is caused by its higher return, which is also quite an important portfolio characteristic for the investor. A weighted grade of membership ($\alpha$) is 0.366. However, a real grade of membership for this solution is 0.415. The highest grade of membership is, of course, for a risk, (0.777). On the other side, the less important objective cost provides the lowest level of membership grade (0.415). This fact is understandable because higher cost does not matter much to the investor.

The risk-averse investor tries further to reduce any risk connected with the investment. The question is how to do it. At first, as the extreme objective values confirm, a risk reduction is possible. Second, it is interesting to find out whether it is possible to do a risk reduction without reducing the return. In many cases on the capital market, such an operation is not possible. However, this is not always the case, especially in a situation of not so value-flexible risk measure. When the portfolio cost is not too important a characteristic, some part of its value can be sacrificed in favor of the other criteria. So, let us try to do this change of the current portfolio. Under the condition of a not deteriorating return, the cost can be relaxed by 0.147 percentage point to reduce a portfolio risk. Then, Model (24) in the following particular form is solved

$$\max \quad \alpha$$

$$(1-0.529)\frac{4.3-\sum_{i=1}^{20}l_i x_i}{2.3} \geq \alpha$$

$$x_i \in X^1 \cup X^2 \quad i = 1,2,...,20 \tag{41}$$

$$0 \leq \alpha \leq 1$$

where the set $X^2$ is specified in the patterns in Models (12) and (27) as

$$X^2 = \left\{ \begin{array}{l} \sum_{i=1}^{20} r_i x_i \geq 0.255 \\ \sum_{i=1}^{20} l_i x_i \leq 2.513 - 10^{-8} \\ \sum_{i=1}^{20} c_i x_i \leq 3.553 + 0.147 \end{array} \right\} . \tag{42}$$

The solution of Model (41) represents a revised portfolio consisting of the same mutual funds as the previous one. However, the shares are different. The portfolio composition is 37.32% *Conseq Corporate Bond A*, 32.68% *Conseq Real Estate*, 15% *Conseq Invest Bond A,* and 15% *Conseq Invest Europe Equity B*. The return of the portfolio is the same (0.255%), the risk is 2.45, and the cost is 3.692%. The weighted grade of the membership of this solution is slightly higher; in contrast, the real grade of membership is lower due to an increased cost. As the input data suggest, it is really possible to reduce the risk while maintaining (or increasing) return. The preference for a return increase would mean a lower risk reduction than a stable level of portfolio return. As expected, the share in the real estate fund increased due to its higher return at the expense of the shares of all other funds in the portfolio.

If further cost increases would be acceptable, the process of improving the risk and return values could continue. However, the area for this change is significantly limited due to a maximum possible level of cost (4.3%). Another way to reduce risk is a reduction at the expanse of return under the same or better level of cost. This path leads through the replacement of the equity fund by a lower-risk fund with significantly lower return. In other way, while the risk is cannot be reduced much, the return can still decrease more significantly, as the extreme objective values show. Thus, such a change is not acceptable by the investor. In these circumstances, s/he decides to accept the last known fund portfolio.

*5.1. Discussion*

The introduced method accepts any investment strategy, e.g., 'risk-seeking' for potentially higher return. Then, the equity funds (generally with the highest risk, allowing a possible higher return) would be in a maximum feasible share in the portfolio. This approach could be applied to making a portfolio from another financial investment instruments (stocks, bonds, investment certificates, etc.). However, it can also be used in another field of practice: project management, production systems, service providing, etc. A partial evidence of the revised KSU-STEM applicability on other data can be found in [32], which confirms the significant support of this method in selecting a portfolio of the funds offered by Česká spořitelna.

A non-traditional multi-objective approach of decision-making theory seems to be a very effective tool for a portfolio making. Compared to the one-criterion (fundamental, technical analysis, etc.), or 'human intuition' concepts, it provides a far more complex view to a portfolio selection. Moreover, the investor knows exactly what part of his budget is invested in what investment instrument. Unlike a well-known mean-variance, or more generally mean-risk, optimization approach, the proposed concept enables considering a wide range of quantitative, or qualitative investment criteria as well as all investor's preferences. It is evident that an application power of the proposed concept grows with an increasing number of the criteria where the solution is very difficult to predict. This fact is eventually confirmed by the presented investment case. Finally, a significant applicability of the improved approach is also supported by an implementation of the selected concepts of fuzzy set theory.

Thus, a real-life case study selecting a portfolio of open unit trusts proves the application power of the proposed revised KSU-STEM method in the field of the capital market. The interactive procedure is very helpful to obtain a satisfactory portfolio composition. The approach is user-friendly. The interactive procedure is simplified by an easier re-setting of the weights. The revised determination of basal objective values avoids finding an unnecessarily pessimistic solution. Sharp conditions (26) or modified form (27) also contribute to more effectively finding a compromise solution. Then, the process is faster, which is also an important aspect today.

## 6. Conclusions

The article deals with a revision of the multiple objective programming method KSU-STEM. An inappropriate technique for the determination of the basal objective value is replaced by a 'less negative' approach. More effective ways for expressing the importance of the objectives throughout a decision-making process are proposed. All algorithmic modifications, or recommendations, lead to more satisfactory applicability, which is demonstrated on a real-life case study selecting portfolio from the CONSEQ open unit trusts traded on the Czech capital market. In general, the article shows a significant usability of the methods of decision-making (and fuzzy set) theory in a portfolio selection problem where other concepts are applied more often. I think it is a shame, which is finally proved in the aforementioned article [32].

In the future research, the proposed revisions may be reviewed on other real or simulated datasets preferably outside the world of investing. The revised KSU-STEM can be also extended by the possibility of expressing often only vague (uncertain) preferences within the interactive procedure by the implementation of other instruments of fuzzy set theory (fuzzy relations).

**Funding:** This research was funded by the Internal Grant Agency of University of Economics, Prague, grant number F4/66/2019.

**Conflicts of Interest:** The author declares no conflict of interest.

**Appendix A**

The father of the fuzzy set theory is Professor L. A. Zadeh, who laid the foundations for this theory through his article [33]. The following basic definitions are taken over from Talašová [34].

**Definition A1.** Let $U$ denote the so-called *universe*. Then, *fuzzy set* $\tilde{A}$ on the universe $U$ is defined by the projection

$$\mu_{\tilde{A}} : U \to \langle 0,1 \rangle . \tag{43}$$

The function $\mu_{\tilde{A}}$ is called a *membership function* of the fuzzy set $\tilde{A}$. For each $x \in U$, the value of $\mu_{\tilde{A}}(x)$ represents a grade of membership of the element $x$ to the fuzzy set $\tilde{A}$. The closer the value of the membership function $\mu_{\tilde{A}}(x)$ 1, the more the element $x$ belongs to the set $\tilde{A}$.

**Definition A2.** Let $\tilde{A}$ be the fuzzy set given on the universe $U$ and $\alpha \in \langle 0,1 \rangle$. Then, the *α-cut* of the fuzzy set $\tilde{A}$ is the strict set

$$A_{\alpha} = \{ x \in U \mid \mu_{\tilde{A}}(x) \geq \alpha \} . \tag{44}$$

**Definition A3.** The fuzzy set $\tilde{A}$ defined on the linear space $U$ is called *convex*, if for each $\alpha \in (0,1\rangle$ the α-cut $A_{\alpha}$ is a convex set, i.e., if for the strict set $A_{\alpha}$, the following holds:

$$\forall x, y \in A_{\alpha} \ \forall \lambda \in \langle 0,1 \rangle : \lambda x + (1-\lambda) y \in A_{\alpha} . \tag{45}$$

**Definition A4.** A *fuzzy number* is such a convex fuzzy set $\tilde{F}$ that it holds
$\exists x_0 \in R, \mu_{\tilde{F}}(x_0) = 1,$
$\mu_{\tilde{F}}(x)$ is a piecewise continuous function.

**Definition A5.** A *triangular fuzzy number* is a fuzzy number $\tilde{F}$ whose membership function $\mu_{\tilde{F}}(x)$ has a triangular shape as follows

$$\mu_{\tilde{F}}(x) = \begin{cases} 0 & x \leq a \wedge x \geq c \\ \frac{x-a}{b-a} & a \leq x \leq b \\ \frac{c-x}{c-b} & b \leq x \leq c \\ 1 & x = b \end{cases} . \tag{46}$$

where $a$, $b$, and $c$ are lower, medium, and upper parameters of the triangular fuzzy number $\tilde{F}$. The triangular fuzzy number can be formally written as $\tilde{F} = (a,b,c)$.

**Bellman's optimality principle** is used for finding a solution of the mathematical models with conditions that may not hold strictly [31]. First, two important notions must be defined. A *fuzzy condition* is such a condition that is expressed only approximately. It means that its relation mark does not hold exactly, but with some toleration. A *fuzzy goal* is actually the objective function expressed as a fuzzy condition where a proximity is declared by its extreme values.

Let us use a model with fuzzy goals $G_j, j = 1,2,...,k$, fuzzy conditions $C_j, j = 1,2,...,k$, and strict conditions creating the set $X$. Goals and conditions actually get to the same level. They fulfill the same role in the model. To each $j$-th fuzzy goal, or each $i$-th fuzzy condition, a membership function $\mu_{G_j}(\mathbf{x})$, or $\mu_{C_j}(\mathbf{x})$ is specified. A fuzzy decision (solution) is defined as a fuzzy set that is a result of the intersection of fuzzy goals and fuzzy conditions. Under the condition of a solution feasibility declared by the strict set $X$, the fuzzy decision can be formulated as a fuzzy set $A$

$$A = G_1 \cap G_2 \cap \ldots \cap G_k \cap C_1 \cap O_2 \cap \ldots \cap C_m \cap X. \tag{47}$$

The membership function of the fuzzy set $A$ is specified through the following operation

$$\mu_A(\mathbf{x}) = \mu_{G_1}(\mathbf{x}) \wedge \mu_{G_2}(\mathbf{x}) \wedge \ldots \wedge \mu_{G_k}(\mathbf{x}) \wedge \mu_{C_1}(\mathbf{x}) \wedge \mu_{C_2}(\mathbf{x}) \wedge \ldots \wedge \mu_{C_m}(\mathbf{x}) = \min_{i,j} \left\{ \mu_{G_j}(\mathbf{x}), \mu_{C_i}(\mathbf{x}) \right\}. \tag{48}$$

Then, the maximizing decision can be defined as a fuzzy number with this membership function

$$\mu_A(\mathbf{x^M}) = \begin{cases} \max \mu_A(\mathbf{x}) & x \in X \\ 0 & x \notin X \end{cases}. \tag{49}$$

If the function $\mu_A(\mathbf{x})$ has one unique maximum $\mathbf{x^M}$, then this solution (decision) is classified as strict (certain), which represents all goals and conditions with the highest possible grade of membership.

It is obvious that the solution is obtained on the basis of the *maximin* operator. Thus, the maximum of the membership function $\mu_A(\mathbf{x})$ in the problem with fuzzy goals and fuzzy conditions can be found through the following supporting strict model [35]

$$\begin{aligned} &\max \; \alpha \\ &\mu_{G_j}(\mathbf{x}) \geq \alpha \quad j = 1, 2, \ldots, k \\ &\mu_{C_i}(\mathbf{x}) \geq \alpha \quad i = 1, 2, \ldots, m, \\ &\mathbf{x} \in X \\ &0 \leq \alpha \leq 1 \end{aligned} \tag{50}$$

when $\mu_A(\mathbf{x}) = \min_{i,j} \left\{ \mu_{G_j}(\mathbf{x}), \mu_{C_i}(\mathbf{x}) \right\}$. The idea of Bellman's approach can be expressed by another formulation of the mathematical model, which can be studied by Gupta and Bhattacharjee [36] and Mohamed [37]. The linear model can be solved by a simplex method. In the case of nonlinearity, a gradient or interior point method could be applied. The (mixed) integer mathematical programming model can be solved using the branch and bound method. Thus, a selection of the suitable method is influenced by variable specifications, objective function characters, or searched extremes.

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
