# Peer review of "Algorithmic Improvements of the KSU-STEM Method Verified on a Fund Portfolio Selection"

_information, doi:10.3390/info11050262_

Round 1

Reviewer 1 Report

In this paper, the author introduces the improved method of the algorithm SKU-STEM, and further applies the improved algorithm in the fund portfolio selection. Some conclusions seem to be useful, but this manuscript can be improved from the following aspects.

  1. The author should further clarify and highlight the motivation and contribution of this paper in the Abstract and Introduction.
  2. In improving the algorithm SKU-STEM, the author applies the knowledge related to fuzzy sets. However, in the introduction, the author does not explain the advantages and functions of fuzzy sets in improving the algorithm. It is suggested that the author describe the reasons for choosing fuzzy sets to improve the algorithm in the Introduction.
  3. In Section 3.2.3, the author compared algorithm STEM with algorithm SKU-STEM. Then, an improved algorithm is proposed. I think it is more reasonable to compare the improved algorithm with the two algorithms STEM and SKU-STEM, or show that the improved algorithm makes up for all the shortcomings of the algorithm SKU-STEM?
  4. The format of references is not uniform. Eg.31
  5. There is almost no recent literature in References. It is suggested that the author should refer to more recent literature related to this topic

Reviewer 2 Report

The author presents an improved method of the SKU -STEM algorithm and applies it in the portfolio selection of a fund. The research is interesting and presents importance to academics and professional portfolio managers. 

The paper is an advance on a previous research made by the author 

The STEM and KSU-STEM has been broadly researched for some time now (as the reference show) and a comparison between the two is provided by the authors, but the novelty of the study is the introduction of the fuzzy sets in the analysis.

An improve could be made in explaining why fuzzy sets and what are the advantages

The conclusions are supported by the results of the study and in line with previous research.

The references present past and recent works on the topic. More recent literature could be added.

In terms of English language, I don’t feel qualified to judge, but the paper is smooth, and I had no problem in understanding it.

I think the paper is well written and the research is well organized and presented and I propose to accept it

Round 2

Reviewer 1 Report

Authors have revised the manuscript according to comments, so I think it can be accepted to be published in the journal.